# Effect of NaCl Replacement by other Salts on the Quality of Bísaro Pork Sausages (PGI Chouriça de Vinhais)

**DOI:** 10.3390/foods10050961

**Published:** 2021-04-28

**Authors:** Alfredo Teixeira, Rubén Domínguez, Iasmin Ferreira, Etelvina Pereira, Leticia Estevinho, Sandra Rodrigues, José M. Lorenzo

**Affiliations:** 1Centro de Investigação de Montanha (CIMO), Instituto Politécnico de Bragança, Campus de Santa Apolónia, 5300-253 Bragança, Portugal; ias.ferreira@hotmail.com (I.F.); etelvina@ipb.pt (E.P.); leticia@ipb.pt (L.E.); srodrigues@ipb.pt (S.R.); 2Centro Tecnológico de la Carne de Galicia, Rua Galicia Nº 4, Parque Tecnológico de Galicia, San Cibrao das Viñas, 32900 Ourense, Spain; rubendominguez@ceteca.net (R.D.); jmlorenzo@ceteca.net (J.M.L.); 3Área de Tecnología de los Alimentos, Facultad de Ciencias de Ourense, Universidad de Vigo, 32004 Ourense, Spain

**Keywords:** KCl, Sub4Salt^®^, pork, reduced-salt sausages, low-salt, meat quality

## Abstract

Concerned about the trend to reduce salt consumption, the meat industry has been increasing the strategies to produce and commercialize products where the reduction or even the replacement of NaCl is an important goal. The aim of this study was to test the effect of partial NaCl replacement by KCl and Sub4Salt^®^ on the quality of pork sausages. Three different formulations (NaCl + KCl, NaCl + Sub4Salt^®^, and KCl + Sub4Salt^®^) were considered and compared to the control (2% NaCl). Physicochemical properties, chemical composition, and microbiological and sensory characteristics were evaluated. The replacement of NaCl did not affect pH, water activity (a_w_) or its chemical composition after eight or 16 days ripening time, while a significant sodium reduction was achieved. The oxidation index expressed in TBARS was also not affected by the NaCl substitution and varied between 0.01 to 0.04 of malonaldehyde (MDA) per kg of sample. Similarly, the NaCl replacement did not change the microbiological quality of the sausages, and the production of healthier meat sausages had also no significant effect on their sensory characteristics. Therefore, according to the results obtained, it is viable and a good strategy for the meat industry to produce “reduced sodium content” sausages without affecting their traditional quality.

## 1. Introduction

In recent decades, consumers’ concerns about meat quality have been increasing, especially in relation to health quality. The amount of salt, particularly NaCl, in processed meat products by its link to cardiovascular diseases has been a great concern of health authorities and consumers. Therefore, the meat industry has been increasing the strategies to produce and commercialize healthier products where the reduction or even elimination of sodium is an important goal. However, the ancestral method for curing meat products with salt is still used today as one of the most effective ways to reduce the water activity, increasing the stability, and improving the shelf life of meat products. The direct reduction of salt is not a viable strategy, since both microbial stability and sausage quality (physicochemical and sensorial) depend on the salt content and the ripening reactions [1]. Therefore, during the last decades, the reduction of salt levels have been achieved by the use of salt substitutes such as other metallic salts and the use off-flavor enhancers such as monosodium glutamate or yeast extract as strategies adopted in meat processing [2,3].

Several studies on pork meat show that sodium chloride could be substituted by other salts, mainly potassium, calcium, and magnesium chlorides, in the manufacturing process of dry cured ham [4,5], pork sausages [6,7,8], pork patties [9], pork lacón [10,11] or Bologna sausages [3]. However, different results were obtained according their impact on the physicochemical properties, microbial stability, and sensory quality. NaCl partial replacement by other chloride salts could affect different physiochemical parameters [11], mineral contents [1], and/or lipid oxidation [10] and could have a strong influence on the sensory properties of the final product [3,10,11]. Mixtures of calcium, magnesium, and potassium chloride were used to partially replace sodium chloride in low-fat mortadella [12] and frankfurter sausages [13]. The results suggested the need to make formulation adjustments to optimize the sensory properties and stability of the emulsion. A good strategy to improve the palatability of reduced salt foods relies on the use of salt-replacing ingredients and the most commonly used is potassium chloride [10]. Moreover, the preservative effect of KCl against pathogenic bacterial strains is comparable to that obtained using NaCl, which confirms that KCl has an equivalent antimicrobial effect [14]. Similar results were obtained by other authors who concluded that the partial replacement of NaCl by other salts in dry-cured ham [15] and the salt reduction in hotdog sausages, bacon, ham, and salami [16] did not affect the growth of spoilage or pathogenic microorganism. Therefore, the correct salt reduction does not result in a major change in the composition of the microbial community [17].

Most traditional pork meat products with quality brands, such as Protected Designation of Origin (PDO) and Protected Geographical Indication (PGI), follow ancestral production procedures using excessive levels of NaCl. The partial or even the total replacement of NaCl by other salts can modify the physicochemical, sensory, and microbiological characteristics of these products, thereby altering their quality. Therefore, knowing that NaCl plays an important role in the manufacture of swine products, it is necessary to know what the impact of sodium chloride substitution would be on the quality and sensory traits of these traditional DOP and PGI products. Thus, the aim of this study was to investigate the effect of NaCl replacement by other salts on the physicochemical, sensory, and microbiological quality of Bísaro pork meat sausages (*Chouriça de Vinhais*, PGI-Protected geographical indication). At the same time, the effect of ripening time was also assessed.

## 2. Materials and Methods

The “Chouriça (or *linguiça*) de Carne de Vinhais” is recognized as a protected geographical indication, such as Chouriça de carne de Vinhais or sausage of Vinhais, by order No. 139/96, registered as a PGI product through Commission Regulation (EC) No 1265/98 [18] of 18 June 1998, under registration PT/PGI/0005/0037.

### 2.1. Sausage Manufacturing and Sampling

The meat sausages (Chouriça de Vinhais) were produced at the Carcass and Meat Technology and Quality Laboratory with Bísaro pork supplied by Bísaro Salsicharia Lda (Bragança, Portugal). It is a product with meat and pork fat from the Bísara breed, filled in pork or beef tripe, cylindrical, and smoked. The meat and fat used are properly seasoned with salt, red or white wine from the region, water, garlic, paprika, and bay leaves. It is horseshoe-shaped, 30 to 35 cm long, and reddish brown in color. Figure 1 shows the flow chart of sausage fabrication.

Sausages were produced according to the specifications book of the PGI product, but some changes related to salt substitution were made. Table 1 shows the formulations of the four types of sausages manufactured. Regarding the traditional formulation identified as the control, NaCl was partially or totally replaced by KCl and Sub4Salt^®^. The Sub4salt is a salt mixture with NaCl, KCl, and sodium gluconate commercially available. This mixture was selected because it allows up to 35% reduction of salt without significant taste differences in sausages [19]. The sausages were analyses in two different dry-ripening times (eight and 16 days).

Two replications of sausages were manufactured at different times from the same batch of ground meat with the same formulation. For each replicated lot, three samples of each sausage type were randomly selected, and each sample was analyzed in triplicate for each physicochemical analysis.

### 2.2. Physicochemical Analysis

The pH measurement was determined according to the Portuguese standard [20] using a Crison 507 pH-meter equipped with a 32–52 puncture electrode. Water activity (a_w_) was assessed according to Association of Official Agricultural Chemists (AOAC) [21] using a HigroPalmAw1 rotronic 8303 probe, Bassersdorf, Switzerland. Portuguese standard procedures were used for the determination of moisture [22], ash [23], protein [24], and total chloride [25] contents. The NaCl content was calculated according to European Regulation [26]. A Minolta CM-2006d spectrophotometer (Konica Minolta Holdings, Inc., Osaka, Japan) in CIEL*a*b* space was used for the sausage color determination. Color was assessed on the external surface at three different points. Lipid oxidation was also determined with the thiobarbituric acid reactive substances (TBARs) index using the spectrophotometric method according to the Portuguese standard [27]. The mineral composition of sausages was assayed following the procedure described by Lorenzo et al. [11]. All parameters were measured in triplicate on the left, central, and right portions of the sausage according to Figure 2.

### 2.3. Microbiological Analysis

The sausages were grounded in a sterile chamber, and a portion of 10 g of the sample was transferred to flasks with 90 mL of peptone water and homogenized. For the following dilutions, 1 mL of the first dilution was transferred to a Falcon tube with 9 mL of peptone water; this process was repeated until the 10^−7^ dilution was obtained.

The counting of aerobic mesophilic bacteria was performed in Petri dishes with fused Plate Count Agar (PCA-DIFCO) medium, which were inoculated, by incorporation, with 1 mL of the 10^−5^, 10^−6^, and 10^−7^ dilutions of sausage dilutions. The incubation was carried out in an oven at 30 °C for 72 h; then, the colonies were counted, and the dilutions were then averaged for each sample; the total count was obtained from the use of Equation (1). The results were expressed in colony-forming units per gram (CFU/g).
(1)CFUg=∑cV∗(n1+0.1n2)∗d
where:

∑c is the sum of the colonies on all counted plates;

*V* is the volume of the inoculum sown in each plate;

*n*_1_ is the number of plates from the first counted dilution;

*n*_2_ is the number of plates from the second counted dilution;

*d* is the dilution from which they were eliminated as first counts.

Mold and yeast counting was performed in Petri dishes with Bengal Rose medium (VWR Chemicals), melted and cooled; 0.1 mL of 10^−2^, 10^−3^, and 10^−4^ dilutions were inoculated by surface spreading, using a sterilized glass spreader. The plates were incubated in an oven at 25 °C for five days, after which the colonies were counted, the dilutions were averaged for each sample, and the total count was obtained using Equation (1). The results were expressed in colony-forming units per gram (CFU/g).

Total coliforms and *Escherichia coli* were counted using the Simplate kit (Merk). Initially, the Simplate medium was hydrated with sterile water, and then 1 mL of 10^−1^ dilution sample was added to each 9 mL of medium, per counting plate. The samples were inoculated into the counting plate, the excess liquid was removed by absorption in cotton, and the plate was incubated in an oven at 37 °C for 24 h. Wells that had changed color were considered positive, and the presence of *E. coli* is identified by fluorescence when the plate was exposed to UV light with a wavelength of 365 nm. The results were expressed in colony forming units per gram (CFU/g).

For sulphite-reducing Clostridia counting, aliquots of 10, 5, 1, and 0.1 mL of the initial suspension were added to a tube kept at 80 °C for 10 min and covered with differential reinforced Clostridial broth (Himedia) and incubated at 37 °C for five days. Then, black colonies were counted; results were expressed in colony forming units per gram (CFU/g).

The detection of *Staphylococcus aureus* was performed according to the Portuguese standard protocol [28]. Serial dilutions of the sample were inoculated in Baird-–Parker broth with egg yolk tellurite (Himedia) at 37 °C for 24 h. Then, 3–5 characteristic colonies were selected to test for the presence of coagulase and catalase with rabbit plasma. Results were expressed as colony-forming units per gram (CFU/g).

To detect the presence of *Salmonella* sp. in the samples, the immunodiffusion 1–2 test was applied [29]. In brief, 25 g of sample was weighed and dissolved in 225 mL of peptone water and incubated at 37 °C for 24 h). The results were interpreted visually by observing the development of an immunoband.

### 2.4. Sensorial Analysis

The dry-ripened sausages were evaluated by a qualified and trained taste panel of 10-panelists, following Portuguese standards [30] and using a 7-point scale, where 1 represents the minimum or absence of sensation and 7 the maximum or excessive value. The evaluated attributes were appearance (external color and internal color of the sausages observed after cutting), odor (intensity before and after cutting), taste (salty, bitter, and metallic), texture (firmness perceived by the thumb and hardness, juiciness, and chewability in the mouth), and the flavor (overall sensation, intensity, and persistence). Five samples of each formulation were evaluated considering salt formulations in two sessions for a shorter maturation time. The same number of samples was evaluated for a longer maturation time. The panelists evaluated a set of four different sausages, with a 30-min interval before evaluating the next set. Panelists were allowed to take their time to make their own evaluation so they did not get saturated/fatigued. The sessions occurred on two different days. On the first day, they evaluated three sets of four different sausages, and in the second, two sets were evaluated. Sausages with different ripening times were evaluated in separate sessions, meaning four sessions in total. Panelists had plain water, unsalted toasts, and a Granny Smith apple to cleanse their palates between each sample.

Sausage samples that had cured for eight days were submitted to thermal processing in an electric grill, with heat above and below, until reaching 80 °C in its thermal center. The 16-day cured sausages did not undergo any heat treatment. Both were evaluated while raw externally and internally after cutting into slices of about 5 mm. Those with a shorter curing time were cut after heat treatment. All samples were coded adequately with random 3-digit numbers and served to the assessors monadically.

### 2.5. Statistical Analysis

A standard least square model was fitted to analyze the differences among the four types of sausages and two ripening times. Physicochemical and microbiological data were analyzed using the statistical package JMP^®^ Pro 13.1.0 by Copyright © 2016 SAS Institute Inc. The predicted means obtained for the effects of salting treatment (*T*), ripening time (*R*), and *T* × *R* interaction were ranked based on pair-wise least significance differences and compared using Tukey’s HSD test for * *p* < 0.05, ** *p* < 0.01 or *** *p* < 0.001 significance levels. Statistical analysis of the sensory data was performed using the XLStat program (Addinsoft, New York, NY, USA), a Microsoft Office Excel add-in. Generalized procrustean analysis (GPA), which minimizes the differences between assessors, identifies agreement between them, and summarizes the sets of 3-dimensional data, was used to develop a sensory profile for the sausages. Data matrices of 4 (types of sausages) by 14 (sensory attributes) for the 10 assessors were matched to find a consensus.

## 3. Results and Discussion

### 3.1. Effect of Salt Treatment and Ripening Time on Physicochemical Properties and Chemical Composition

The results of the physical and color parameters are shown in Table 2. No statistical differences were found between salt treatments for pH that varied between 5.7 and 5.9. In a previous study characterizing commercial samples of Chouriça de Vinhais, the pH value was 5.4, which is below that found in the tested salt formulations [31]. In our study, the NaCl replacement did not affect pH. Likewise, several authors observed that in other dry sausages, such as Harbin, 30% replacement by KCl did not produce any pH variation [32,33], and the values ranged between 5.7–5.9 on days 6, 9, and 12. In similar way, the 50% NaCl replacement (by KCl or a combination of potassium, magnesium, and calcium chlorides) in Italian salami sausages [17] or 16% [34] and 28% [6] NaCl replacement by KCl in dry-fermented sausages did not influence the pH. The same trend was also observed in the meat products made from whole pieces, such as lacón [1,11], or in other processed meat products [16], where the salt substitution did not affect the pH. In contrary, other researchers found that the use of salt substitutes with CaCl_2_ produced a significant reduction in pH [13]. In our study, the ripening time (between eight and 16 days) did not affect the pH values, which agrees with the data reported for Harbin sausages [32], where no significant differences were observed in sausage pH between days 9 and 12, although a dramatic pH reduction was detected during the first six days of fermentation.

On the other hand, ripening time had a strong influence on the a_w_ value, while the salt treatment had a very low effect on this parameter. The a_w_ values of sausages ripened for eight days were around 0.91–0.93 and for 16 days varied from 0.85 to 0.89. These results were lower than the a_w_ recorded for sal-reduced hot dogs [8]. Some Spanish dry-fermented sausage commercial brands also have lower a_w_ values (0.648–0.894) [35] if compared with the sausages ripened for eight days. However, the comparison between different studies using NaCl replacements is difficult, since the a_w_ values are highly dependent on the drying step and are modulated by both, formulation (fat amount, other salts, and ingredients or spices, etc.), and processing conditions (temperature, relative humidity, air velocity, etc.). In the present study, the ripening time had an important impact on a_w_. The sausages ripened for 16 days presented significantly lower a_w_ values than those ripened for 8 days. Several authors reported the same behavior found in the present study, since the a_w_ (and moisture content) decreased as the dry-ripening process increased [32,33]. Additionally, in our case, the sausages ripened for 16 days show higher stability, since high a_w_ and moisture contents result in highly perishable food. In fact, in a previous study, the authors confirmed that about 40% of Vinhais sausages presented a_w_ values higher than 0.90 [36].

Regarding the effect of salt treatment, the a_w_ values were similar between the treatments, with no statistical difference found in relation to the control in the ripening period of eight days, while in the sausages ripened for 16 days, the treatment KCl + Sub4Salt^®^ showed higher a_w_ values in relation to the other reformulated samples, but not to the control. Similarly, other authors also reported that the a_w_ value was not affected by the partial replacement of NaCl in dry sausages [17,33,34]. By contrast, in dry-fermented sausages [6], 28% replacement of NaCl by KCl resulted in a slight (significantly) increase in a_w_. Further, in dry Harbin sausage, partial NaCl replacement (30%) resulted in higher a_w_ values [32]. This fact was attributed to the lower capacity of reducing this parameter by KCl compared to NaCl [32].

The instrumental color coordinates showed that the NaCl replacement only influenced the sausage luminosity (L*), while ripening time changed all color parameters (Table 2). In sausages from the treatments that used sub4Salt as salt replacers, there was a significant L* increase after eight days of the ripening process, while in all reformulated samples, this parameter was lower than in control samples after 16 ripening days. In salami, depending on the salt substitution formulation, it was also observed that the partial replacement increased L* and b* parameters, while the redness presented variable behavior [37]. In accordance with our results, the low influence of salt replacement was also observed by several authors, who found that partial NaCl did not have or had a minor effect on color parameters [6,12,32]. Additionally, the minimal influence of NaCl replacement on color coordinates were classified as “no important difference”, since it may not be discernable by human vision [38]. Similar results were also reported for lacón, where the authors observed that the partial replacement of NaCl by KCl or two different combinations of KCl, MgCL_2_, and CaCl_2_ did not influence color parameters in any manufacturing step, while as the ripening time progressed, a significant decrease in L* and b* and increase in a* values was found [11]. These findings agree with the results obtained in the present research, where the ripening process decreased both L* and b* values in the control and reformulated sausages. In the case of a*, in control samples, this value increased, while decreased in reformulated sausages. These differences in color parameters between different ripening days are related to meat products becoming darker due to weight loss [11]. In ripened meat products, this weight loss is mainly due to the moisture loss during the drying process. Thus, the lowest values of these parameters in 16-day ripened in comparison with 8-day ripened sausages are expected, since 16-day ripened sausages had significantly lower moisture contents (Table 3).

The proximate composition and lipid oxidation (TBARs) of different sausage formulations are shown in Table 3. As the ripening step progressed, the fat and protein contents increases significantly, while the moisture content decreased. Similarly, the ash content also increased, although this change was not significant. This is because the progressive loss of moisture during ripening causes an increase in the dry matter (which includes protein, fat, and ash). A similar trend was observed in dry-ripened sausages [39]. These authors found in chorizo an increase in the content of lipids from about 32% immediately after production to 45–60% after the curing process (35 days). Although in the present study, the sausages were not evaluated immediately after production, it was possible to identify the same tendency to increase the fat (and protein) content between eight and 16 days of ripening, while the formulations used in the present study showed a proximate composition compatible with the sausages found commercially [31].

The protein content found in this study was 26.2–27.4% in 8-day ripened and 32.4–37.0% in 16-day ripened sausages. These data are similar to those reported for traditional Portuguese sausages (about 24.5%) [40]. In another study characterizing sausages from PGI Vinhais, the protein content ranged from 12.3% to 35.9% [31], which agrees with the results obtained in the present research. The salt formulations did not affect the protein percentage of sausages ripened for eight days. However, in the sausages ripened for 16 days, the protein content was significantly higher in the two treatments with Sub4Salt than the control or NaCl + KCl formulations.

Regarding the fat content, our values ranged from 15.6 to 22.2%. This content is lower than those reported in previous study on commercial “Chouriza de Vinhais”, where the fat content varied between 20.4 and 68.2% [31]. However, the variation between studies may be due to the different samples manufactured, which vary their formulation, with a lack of homogeneity of the processing methods used. The fat content was significantly affected by salt treatment. Generally speaking, the formulations incorporating Sub4Salt presented the lowest fat content for both ripening periods. This is the opposite behavior previously mentioned for the protein. Nevertheless, it is important to highlight that the variations between control and reformulated sausages were not significant, since the main differences were between NaCl + KCl and the two formulations that included Sub4Salt.

For the moisture content, the values ranged from 33.9–36.2% in 8-day ripened to 44.9–49.6% in 16-day ripened sausages. These contents agree with those reported in a study of different types of Portuguese traditional sausages [40]. The average moisture found by these authors was 32.5%, a value that is close to the average for the 16-day ripened sausages (35.1%). Our moisture values showed significant differences both in relation to the salt treatment and the ripening time. As mentioned above, and in accordance with other authors, during the ripening period, the moisture decreased constantly as a consequence of temperature and humidity conditions in the dry-curing or dry-ripening step [11,32,39]. Regarding the salt treatment effect, in 8-day ripened sausages, formulations containing Sub4Salt as the NaCl replacer presented the highest moisture values, while in 16-day samples, the control and NaCl + Sub4salt samples had the highest values, and the reformulations with highest KCl amounts (NaCl + KCl and KCl + Sub4Salt) presented the lowest moisture contents. In Harbin sausages, the sausages reformulated with a mixture of KCl and other salts presented a higher moisture content in comparison with the control and sausages reformulated with NaCl + KCl [33]. These results confirm that the interaction between salts and meat proteins could have an effect on the drying process. In fact, a study affirmed that the differences in the dehydration process of reformulated lacón with salt mixtures could be due to the faster penetration of the salt mixtures containing KCl that would make it difficult for water to escape from the interior of the meat [11].

Neither salt formulations nor the maturation period affected the ash content, which ranged from 4.5 to 6.3%. Our values were higher than those reported by other authors for frankfurter sausages (3.6%) [13]. In contrast, Stanley et al. [9] found that the overall proximate composition of pork sausages with sodium chloride replacement by potassium chloride only varied in ash content, and they did not find differences in protein, moisture, or fat content. Further, in mortadella, it was found that the ash content of the formulation without blends of calcium, magnesium, and potassium chloride was significantly lower than that of the others [12].

Finally, in contrast to our findings on the proximate composition, the partial replacement of NaCl by KCl or combination of KCl, MgCl_2,_ and CaCl_2_ did not influence the proximate composition of dry-fermented sausage [34] and Italian salami [17]. In dry-cured lacón, the authors also observed that reformulation had no effect on the protein or fat content, while the moisture presented variable behavior [4].

The chloride content of the sausages is directly related to the treatments used, where there was a change in the proportion of salts used confirmed by the statistically significant difference between all treatments. Further, statistical differences between the ripening period was found, and 8-day ripened sausages had lower chlorides content than the 16-day ripened sausages as result of the moisture reduction. However, chloride determination is not a suitable technique to observe a reduction in the NaCl content, since the rest of the salts also contain the chloride anion. In fact, when calculating the percentage of NaCl with the sodium content (as specified by the European regulation [41]), a significant decrease in NaCl could be observed when comparing the different treatments. As the ripening time increased, the content of NaCl also increased due to the decrease in moisture. Moreover, the NaCl + KCl and KCl + Sub4Salt treatments presented significantly lower NaCl values in comparison with the control, while NaCl + Sub4Salt also had lower values but did not differ significantly from the control sausages. An important salt reduction was achieved in the NaCl + KCl (about 16% reduction for both ripening periods) and KCl + Sub4Salt treatments (20% and 25% reduction in 8- and 16-day ripened sausages, respectively). This was expected since NaCl was replaced by other chloride salts; thus, the final content of NaCl is highly dependent on the NaCl replaced in the formulation. Our findings agree with those reported for Harbin sausage [33], where the reformulation dramatically decreased the NaCl content in the final meat product.

The TBARs value is related to the malondialdehyde (MDA) content, which is one of the main compounds derived from the degradation of lipids and used as an indicator of the degree of lipid oxidation [42]. The control of lipid oxidation is important for meat industry, since this reaction is one of the main causes of the loss of quality of meat products in general and pork sausages in particular [42]. In the present research, the salt formulation or the ripening time did not affect the TBARs value, which varied between 0.01 and 0.04 mg of malonaldehyde (MDA) per kg of sausage. These values are really low and confirm that both control and reformulated sausages maintained the quality, and they did not develop rancid flavor. The typical evolution of TBARs in meat products during ripening or curing stages is an initial and slow increase in this value (initiation phase: this reaction needs some free radicals), followed by an exponential increase (propagation phase) and a posterior decrease because the malondialdehyde is further degraded by other compounds such as volatiles (termination phase) [42]. This behavior is commonly reported by other authors for multiple meat products [10,33,37,43]. However, in this case, it was not possible see this trend, which is related to the high quality of initial meat and fat used in the manufacturing, as well as the correct relationship between anti- and pro-oxidant compounds in the sausage formulation, which implies the delay of lipid oxidation in this product. Additionally, NaCl is recognized as a potent pro-oxidant [42], but in our study, no effect was observed when this salt was partially replaced by others. Similar behavior was observed in Harbin sausages [33], lacón [1,10], and bacon [43] where NaCl replacement by a NaCl + KCl mixture did not affect the TBARs index, in which the use of KCl as a NaCl replacer did not increase or decreased the hexanal content, the main lipid derived-volatile and a good indicator of lipid oxidation. In contrast, several authors observed differences in lipid oxidation when meat products were reformulated with other salts. For example, the replacement of NaCl by a combination of KCl with MgCl_2_ and/or CaCl_2_ significantly increased the TBARs values in Italian salami [17], dry-fermented sausages [37], and bacon [41]. This fact is related to the use of divalent cations (such as Mg^+2^ and Ca^+2^) in the reformulation of meat products, which promotes the lipid oxidation reactions [10,17]. With all these in mind, generally speaking, the reformulation with KCl did not increase lipid oxidation, but when the replacement includes calcium or magnesium chloride salts, special attention must be paid to this phenomenon to avoid a reduction in product quality.

The mineral content of different sausage formulations and ripening times is shown in Table 4. In relation to the mineral content, as predicted, 16-day ripened sausages had a higher mineral content than the 8-day ripened sausages with the exception of the Mn contents. This fact is related, as explained in other sections, to moisture decrease during ripening, which resulted in a higher concentration of components from dry matter. This finding agrees with those previously reported for dry-cured lacón [11]. Similarly, as expected, a significant reduction in the Na content was observed in all reformulated samples in relation to the control and was particularly higher in the KCl + Sub4Salt^®^ sausages.

In our study, the 8-day ripened samples had a Na reduction between 10–20%, and the sausages ripened for 16 days presented values between 8 and 25% less Na than the control. Obviously, the highest reductions were achieved with the formulation KCl + Sub4Salt.

This evidence agrees with the results reported by several authors for other meat products. NaCl replacement by KCl, MgCl_2,_ and/or CaCl_2_ reduced the sodium content in Italian salami by approximately 50% [17]. Additionally, the use of magnesium and calcium chlorides could achieve more than 75% Na reduction [1,11], but this depends on the NaCl reduction in the initial meat product formulation. In agreement with other authors, the partial substitution of sodium chloride by the mixture of chloride salts not only reduced Na but significantly increased the potassium content [1,11,32]. Moreover, the use of blends of chloride salts (with magnesium and calcium salts) as partial substitutes for NaCl favored a balanced intake of minerals in frankfurter sausages [13], Italian salami [17], and lacón [1,11] since in these meat products, a dramatic increase in Ca and Mg contents was observed, and these are important minerals for maintaining adequate health. In this research, the reformulation did not increase Ca because all salts added in the sausage formulation contain this mineral. Marcos et al. [40] found an average value of 30 mg of Ca/100 g for Portuguese traditional meat sausages, which is close to the value obtained in our control formulation and the other formulations containing sodium as NaCl + KCl and NaCl + Sub4Salt^®^, with 30.2 and 28.6 mg/100 g average values of the 8- and 16-day ripened sausages, respectively.

The NaCl + KCl formulations unexpectedly presented the highest K content, and the control formulation, despite presenting the lowest value of K, had a relatively high values. This fact may be due to the use of paprika dry paste (*Capsicum annuum*) used in the traditional formulation of this sausage, which has considerable values of minerals, mainly potassium [44]. Although the ripening time improved the Fe, P, and Zn contents, no significant differences were found between the salt formulations. Similarly, in Italian salami, salt replacement by KCl + MgCl_2_ + CaCl_2_ did not influence the P content [17]. In our study, the P content varied between 295.2 and 381.8 mg/100 g, higher contents that those reported for Italian salami (261–286 mg/100 g) [17]. In contrast, a slightly higher P content was found in traditional Spanish sausages than in our sausages; the average content of this mineral was 431 mg/100 g [44]. The paprika paste also interferes with the phosphorus content, since it has a significant amount of this mineral. Finally, the Zn contents are within the range of the amount of Zn found in meat, ranging from 1.6 to 4 mg/100 g [45]. The values of zinc present are also in accordance with data reported for a traditional Spanish dry sausage (7.4 mg/100 g) or for a dry common sausage (3 mg/100 g) [44].

### 3.2. Effect of Salt Treatment and Ripening Time on Microbiological Counts

The limit established in Portugal for total aerobic microorganisms in sausages is 5 × 10^6^ CFU/g (6.7 log CFU/g). Regarding *E. coli*, the limit defined by the Commission Regulation [46] is 5 × 10^3^ CFU/g (3.7 log CFU/g). The counts of total aerobic microorganisms at 30 °C and of molds and yeasts are presented in Table 5. It is important to highlight that sulphite-reducing Clostridia, *S. aureus*, *Salmonella* sp., were not detected in any sample.

The count of total microorganisms at 30 °C ranged from 6.51 to 7.31 log CFU/g; no significant differences between treatments, curing time or between the interaction of both parameters were observed. In a study on Chouriça of Vinhais, values varied between 5.1 and >7.5 log CFU/g [36], which include those found in the present study and can be attributed to the fermentation process and the growth of lactic acid bacteria over the ripening time. Similarly, other authors who also studied the microbiological profile of Chouriça de Vinhais found, at the end of the processing, values for the count of total microorganisms ranging from 8 to 9.5 log CFU/g [47]. These values are above those obtained in the present study, which can be attributed to the different conditions of processing such as time and temperature, raw materials, and the manufacturing process hygiene.

In the formulations evaluated in the present research, values for molds and yeasts ranged from 3.53 to 4.49 log CFU/g. The values obtained are in agreement with another study, in which the mold and yeast count in Chouriça de Vinhais varied between <1 and 5.7 log CFU/g [47]. Contrary to our results, in dry-cured lacón, the yeast counts significantly increased during the ripening stage [11]. However, it is also important to highlight that the manufacturing process in lacón is much longer than that in Chouriça de Vinhais sausage, which could have a strong influence in the development of yeasts.

Our findings did not show significant differences in microbial counts between the different salt formulations. This agrees with the evidence reported by other authors for dry-cured ham in which microbial counts were not influenced by the addition of different chloride salts in their salting stage [4,15]. In Harbin sausages, the authors observed a continuous increase in lactic acid bacteria (during 12 days) and of total aerobic bacteria (first six days), while the 30% NaCl replacement by KCl also did not affect the microbial counts [32]. Similarly, the effects of NaCl replacement by KCl on the bacterial community composition of traditional Chinese bacon was also limited [43]. However, contrary to our results, in dry-cured lacón showed differences between salt formulations [11]. In this case, the batch salted using a mixture of NaCl and KCl at 50% presented the highest total viable counts, salt-tolerant flora, and yeast counts in comparison with the control [11].

Table 5 presents the counts of total coliforms and *E. coli*, according to the replica, ripening time, and salt treatment. A significant difference was found between the manufacturing replicas, where the sausages produced in the first batch had a higher total coliform count, ranging from <1 to 2.89 log UFC/g, and the second batch showed values up to 1.3 log UFC/g. Concerning *E. coli*, all results were <1 log UFC/g. Ferreira et al. [47] obtained similar results for the Vinhais sausages. However, these authors stated that sausages from two producers had values of 5.2 and 5.3 log UFC/g [47]. The *E. coli* counts for the sausages produced and evaluated in the study are classified as satisfactory according to the regulation in force [46] since they are below the stipulated limit (5 × 10^5^ UFC/g to 5 × 10^6^ UFC/g).

### 3.3. Effect of Salt Treatment and Ripening Time on Sensorial Properties

No research related to the study of the sensory properties of reformulated Chouriça de Vinhais using consumer tests was found. In the present research, in order to minimize the differences between assessors, GPA was used to find a consensus. Figure 3 shows the biplot of the consensus configuration with the correlations between sensory attributes, GPA factors F1 and F2 and the coordinates of the different sausages with eight (Figure 3a) and 16 (Figure 3b) days of ripening. Three factors explained all data variability; F1 and F2 together explained 78% and 76% of the total variability in sausages cured for eight and 16 days, respectively.

According to the coordinates of the sausage types (8-day ripened sausages) and the correlation of the sensory attributes with the main factors, it can be said that the control samples had higher interior odor intensity and flavor persistence than reformulated sausages. In the case of NaCl + KCl sausages, they had a dark interior color (darker red), high odor intensity, hardness, and bitter taste. KCl + Sub4Salt sausages had the most noticeable basic taste values, though not very high, and NaCl + Sub4Salt sausages had the highest texture, external color, and flavor intensity.

In sausages with a longer ripening time (16 days), GPA indicated that the control samples had the highest hardness, chewability, exterior color, internal odor, bitter taste, and persistence of flavor. NaCl + KCl samples were considered the juiciest and salty, KCl + Sub4Salt was the one that had the highest firmness, and NaCl + Sub4Salt had the highest intensity of interior color and metallic taste.

In Italian salami, NaCl replacement resulted in a lower global acceptability, partially related to the lower scores or saltiness and color intensity in reformulated samples [17]. In a similar way, the replacement of NaCl by KCl had negative consequences on the aroma, taste, and acceptability of traditional Chinese bacon [43]. Further, in cooked hams, Tamm et al. [48] found that salt reduction by a combined approach of high-pressure treatment and the salt replacer KCl could potentially reduce product acceptability since a lower salty taste was detectable in sensory evaluations. Moreover, in Chinese bacon [43], a gradual and proportional bitter taste was observed with the inclusion of KCl, which agrees with the results reported for Harbin sausages [32] and lacón [10,11]. In contrast, the partial replacement of NaCl by KCl or a mixture of KCl, MgCl_2,_ and/or CaCl_2_ did not influence the sensory quality of restructured bacon [41]. Therefore, taking this into account, it seems clear that sensory analysis of the present research must be expanded in future research, including consumer analysis, in order to test the influence that salt reformulation may have on the acceptability of Chouriça de Vinhais sausages.

Globally, the results of the effect of NaCl substitution on physicochemical, sensory, and microbiological quality in Chouriças de Vinhais sausages clearly indicate that sodium can be completely replaced by KCl and Sub4Salt^®^. Future studies in an industrial environment and consumers’ sensory tests [49] should be carried out in order to obtain healthier meat products.

## 4. Conclusions

The results indicate that the partial replacement of NaCl by KCl or its mixture with Sub4Salt^®^ in Bísaro pork sausage formulation (PGI-Chouriça de Vinhais) did not affect or had only a minor impact on the physicochemical characteristics and chemical composition of the final product. In contrast, as expected, the ripening time strongly influenced these sausage parameters. The oxidation index expressed as TBARs was also not affected by the NaCl substitution or ripening time. An important reduction in sodium and NaCl contents was achieved in sausages reformulated with KCl + Sub4Salt and NaCl + KCl mixtures, resulting in a 22.7% and 16% reduction in the sodium content, respectively. These sausages can be labeled as “reduced sodium content”, which offer the meat industry an opportunity to put on the market a healthy product, demanded by the consumer. Another important fact is that the microbiological quality of the sausages was not affected by the NaCl substitution. Finally, although assessors can distinguish chorizo sausages with different formulations, the results obtained indicate that assessors do not detect very significant differences in sensory characteristics when changing salt levels in sausage formulations, which may be important in the production of healthier products.

Therefore, as a general conclusion, the use of KCl and Sub4Salt in the reformulation of traditional Portuguese sausages showed promising results, with a low impact on typical characteristics, while a significant reduction in NaCl was achieved. However, these results should be probed with additional analysis and completed with a consumer acceptability test.

## Figures and Tables

**Figure 1 foods-10-00961-f001:**
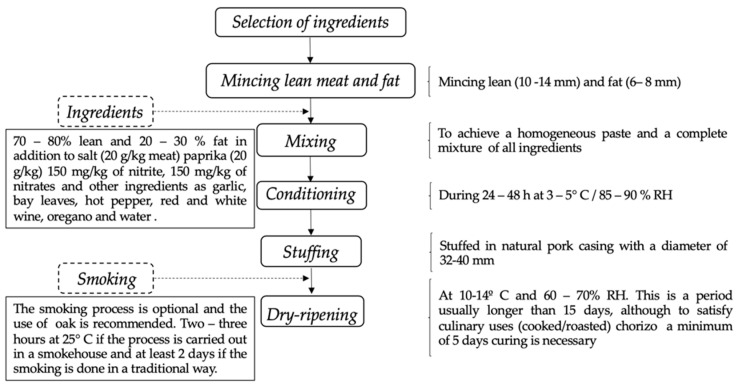
Flow chart of Chouriça de Vinhais sausage fabrication (According to the PGI specifications [18]).

**Figure 2 foods-10-00961-f002:**
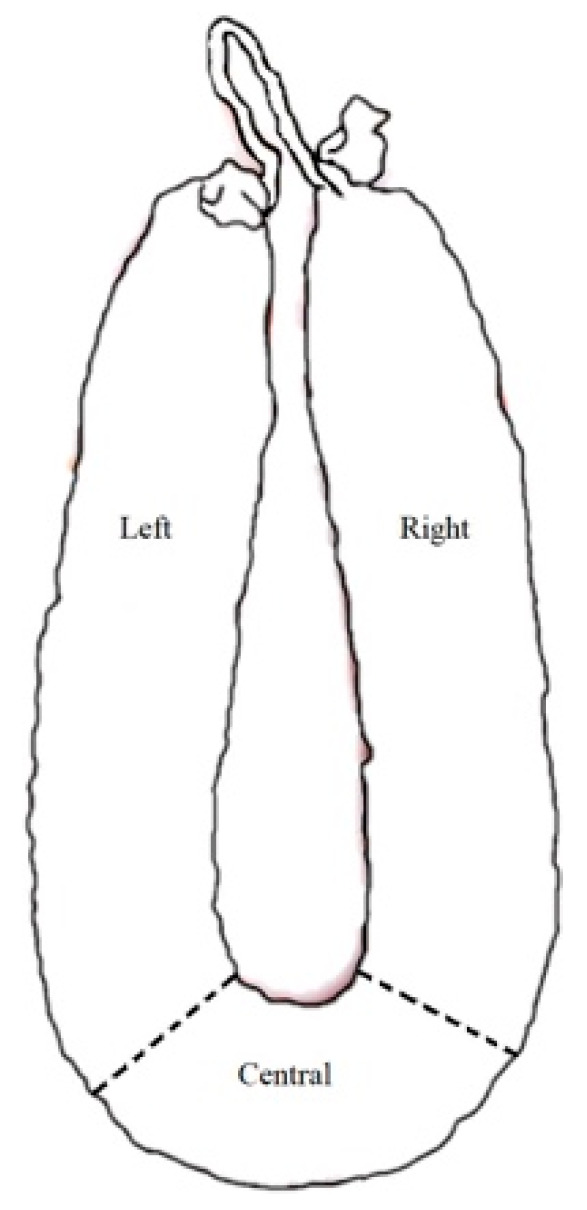
Representation of sausage sampling for physicochemical analysis.

**Figure 3 foods-10-00961-f003:**
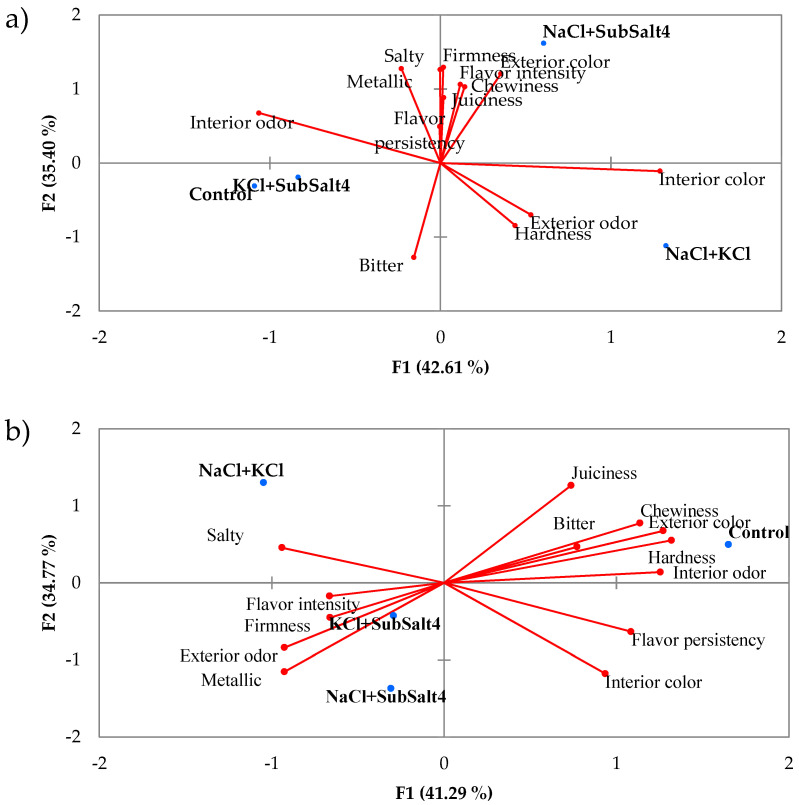
Consensus configuration: joint representation of the correlation between sensory attributes, F1 and F2 and coordinates of the sausages with eight (**a**) and 16 (**b**) days of ripening.

**Table 1 foods-10-00961-t001:** Formulations of the four types of Chouriça de Vinhais.

Formulation	Treatment (*T*)			
	Control	NaCl + KCl	KCl + Sub4Salt^®^	NaCl + Sub4Salt^®^
**Meat**	92.60%	92.60%	92.60%	92.60%
**Spices**	3.40%	3.40%	3.40%	3.40%
**H_2_O**	2%	2%	2%	2%
**NaCl**	2%	1.50%	-	1.50%
**KCl**	-	0.50%	1.50%	-
**Sub4Salt^®^**	-	-	0.50%	0.50%

Meat: 70–80% lean and 20–30% fat; Spices: paprika (20 g/kg) nitrite (150 mg/kg), nitrate (150 mg/kg), garlic, bay leaves, hot pepper, red and white wine, and oregano.

**Table 2 foods-10-00961-t002:** Effect of NaCl replacement by other salts on physical and color parameters under different dry-ripening days of the Chouriça de Vinhais sausages.

Parameter	Salt Treatment (*T*)								SEM	Significance		
	Control		NaCl + KCl		KCl + Sub4Salt		NaCl + Sub4Salt					
	Ripening Time (*R*)									*T*	*R*	*T* × *R*
	8 Days	16 Days	8 Days	16 Days	8 Days	16 Days	8 Days	16 Days				
**pH**	5.9	5.9	5.8	5.9	5.7	5.8	5.8	5.9	0.07	ns	ns	ns
**a_w_**	0.92 ^a^	0.87 ^cd^	0.92 ^a^	0.85 ^d^	0.91 ^ab^	0.89 ^bc^	0.93 ^a^	0.86 ^d^	0.01	ns	***	ns
**Color Parameters**												
**L***	40.9 ^b^	37.6 _c_	39.6 ^b^	33.9 ^d^	45.2 ^a^	35.2 ^d^	43.8 ^a^	33.3 ^d^	0.7	***	***	***
**a***	14.6 ^bc^	16.0 ^ab^	16.1 ^ab^	14.1 ^c^	17.2 ^a^	14.4 ^bc^	17.2 ^a^	14.7 ^bc^	0.61	ns	***	**
**b***	22.0 ^ab^	19.6 ^c^	20.0 ^bc^	16.7 ^d^	23.2 ^a^	18.1 ^cd^	23.2 ^a^	16.9 ^d^	0.84	ns	***	ns

SEM: Standard Error of the Mean; Significance: ** *p* < 0.01; *** *p* < 0.001; ns: not significant; ^a–d^ Mean values in the same row (corresponding to the same parameter) not followed by a common letter differ significantly (*p* < 0.05; Tukey test).

**Table 3 foods-10-00961-t003:** Effect of NaCl replacement by other salts on the proximate composition and lipid oxidation under different dry-ripening days of the Chouriça de Vinhais sausages.

Parameter	Salt Treatment (*T*)								SEM	*p* Value		
	Control		NaCl + KCl		KCl + Sub4Salt		NaCl + Sub4Salt					
	Ripening Time (*R*)									*T*	*R*	*T* × *R*
	8 Days	16 Days	8 Days	16 Days	8 Days	16 Days	8 Days	16 Days				
Proximate Composition (%)												
**Protein**	27.3 ^d^	32.4 ^c^	26.7 ^d^	32.5 ^c^	26.2 ^d^	33.8 ^b^	27.4 ^d^	37.0 ^a^	0.39	***	***	***
**Fat**	16.6 ^bc^	20.6 ^ab^	19.7 ^ab^	22.2 ^a^	16.9 ^bc^	17.3 ^bc^	15.6 ^c^	17.4 ^bc^	1.39	**	*	ns
**Moisture**	45.5 ^b^	35.9 ^c^	44.9 ^b^	33.9 ^d^	48.6 ^a^	34.5 ^d^	49.6 ^a^	36.2 ^c^	0.43	***	***	***
**Ashes**	4.9	5.4	6.3	5.2	4.5	5.9	4.5	6.2	0.57	ns	ns	ns
**Chlorides (%)**	3.5 ^d^	4.4 ^a^	3.4 ^de^	4.2 ^b^	3.1 ^f^	3.8 ^c^	3.3 ^ef^	4.1 ^b^	0.06	***	***	ns
**NaCl (%)**	3.43 ^bc^	4.32 ^a^	2.87 ^d^	3.64 ^bc^	2.74 ^d^	3.25 ^cd^	3.11 ^cd^	3.96 ^ab^	0.11	***	***	ns
**TBARs** (g of (MDA)/kg)	0.03	0.01	0.05	0.01	0.02	0.04	0.02	0.01	0.00	ns	ns	ns

SEM: Standard Error of the Mean; Significance: * *p* < 0.05; ** *p* < 0.01; *** *p* < 0.001; ns: not significant; ^a–f^ Mean values in the same row (corresponding to the same parameter) not followed by a common letter differ significantly (*p* < 0.05; Tukey test).

**Table 4 foods-10-00961-t004:** Effect of NaCl replacement by other salts on mineral contents under different dry-ripening days of the Chouriça de Vinhais sausages.

Parameter	Salt Treatment (*T*)								SEM	*p* Value		
	Control		NaCl + KCl		KCl + Sub4Salt		NaCl + Sub4Salt					
	Ripening Time (*R*)									*T*	*R*	*T* × *R*
	8 Days	16 Days	8 Days	16 Days	8 Days	16 Days	8 Days	16 Days				
Minerals (mg/100 g)												
**Ca**	28.0 ^bc^	41.7 ^a^	27.2 ^bc^	33.3 ^b^	22.2 ^c^	27.7 ^bc^	24.6 ^c^	32.5 ^b^	2.22	*	***	ns
**K**	470.8 ^e^	531.0 ^e^	788.2 ^b^	940.3 ^a^	637.4 ^d^	735.3 ^bc^	480.6 ^e^	665.1 ^cd^	27.3	***	***	ns
**Fe**	1.3 ^b^	1.7 ^a^	1.3 ^b^	1.8 ^a^	1.2 ^b^	1.7 ^a^	1.2 ^b^	1.7 ^a^	0.07	ns	***	ns
**Mn**	40.0	51.7	42.9	48.9	43.4	46.7	42.3	50.4	4.51	ns	ns	ns
**Na**	1372 ^cd^	1727 ^a^	1147 ^ef^	1456 ^bc^	1097 ^f^	1298 ^d^	1243 ^de^	1585 ^b^	45.7	***	***	ns
**P**	295.1 ^c^	380.2 ^a^	310.0 ^c^	381.8 ^a^	302.8 ^c^	349.7 ^b^	298.2 ^c^	375.5 ^ab^	9.39	ns	***	ns
**Zn**	3.7 ^b^	4.7 ^a^	3.7 ^b^	4.7 ^a^	4.8 ^a^	3.6 ^b^	3.6 ^b^	4.5 ^a^	0.13	ns	***	ns

SEM: Standard Error of the Mean; Significance: * *p* < 0.05; *** *p* < 0.001; ns: not significant; ^a–f^ Mean values in the same row (corresponding to the same parameter) not followed by a common letter differ significantly (*p* < 0.05; Tukey test).

**Table 5 foods-10-00961-t005:** Effect of NaCl replacement by other salts on microbiological counts (log CFU/g) under different dry-ripening days of the Chouriça de Vinhais sausage.

Parameter	Salt Treatment (*T*)								SEM	*p* Value		
	Control		NaCl + KCl		KCl + SubSalt4		NaCl + SubSalt4					
	Ripening Time (*R*)									*T*	*R*	*T* × *R*
	8 Days	16 Days	8 Days	16 Days	8 Days	16 Days	8 Days	16 Days				
**Totals**	6.81	7.0	6.84	7.31	6.7	6.82	7.15	6.51	0.24	ns	ns	ns
**Molds and Yeasts**	4.49	3.83	4.15	3.53	3.83	3.74	4.10	3.77	0.14	ns	ns	ns
**Coliforms totals R1 ^†^**	1.3 ^a^	2.89 ^a^	2.24 ^a^	1.30 ^a^	2.26 ^a^	<1 ^a^	1.30 ^a^	2.54 ^a^	0.67	ns	ns	ns
***E. coli* R1 ^†^**	<1	<1	<1	<1	<1	<1	<1	<1	NE	ns	ns	ns
**Coliforms totals R2 ^‡^**	1.3 ^a^	<1 ^b^	<1 ^b^	0 ^b^	0 ^b^	0 ^b^	0 ^b^	0 ^b^	NE	ns	ns	ns
***E. coli* R2 ^‡^**	<1	<1	<1	<1	<1	<1	<1	<1	NE	ns	ns	ns

SEM: Standard Error of the Mean; ns: not significant; ^a,b^ Mean values in the same row (corresponding to the same parameter) not followed by a common letter differ significantly (*p* < 0.05; Tukey test); NE: Not estimable; R1 ^†^: Replica 1 of sausages manufacture; R2 ^‡^: Replica 2 of sausage manufacturing.

## Data Availability

The data presented in this study are available in the article.

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
