# Peer review of "Effect of NaCl Replacement by other Salts on the Quality of Bísaro Pork Sausages (PGI Chouriça de Vinhais)"

_foods, 2021, doi:10.3390/foods10050961_

Round 1

Reviewer 1 Report

The manuscript studied the effect of two salt replacers, KCl and Sub4Salt®, on the quality of a dry sausage typically produced in Portugal. The aim as indicated in lines 77-80 was to investigate the effect of NaCl  replacement by other salts on physicochemical, sensory and microbiological quality of Bísaro pork meat sausages (Chouriça de Vinhais, PGI - Protected geographical indication). At the same time, the effect of ripening time was also assessed. The manuscript shows the changes in pH, Aw, colour, composition, salt content, oxidation value in TBARS values, mineral content and sensory properties. Furthermore, the introduction includes a long description of studies regarding the effect of salt reduction and substitution in meat products. This shows that the novelty of the manuscript is scarce and many studies have already confirmed the efficacy of KCl and other salts in meat product reformulation.

The manuscript includes 70 references. In order to see the relevance of the manuscript only those references directly related with the study should be included. In my opinion, in order to demonstrate the novelty and the results obtained, I consider that the manuscript should include a maximum number of references of 30 to 40.  The authors should be aware of shorten the manuscript and only show the relevant results.

Other comments

Lines 129-131- Please include the type of methodology used not only the reference.

Line 181 Sensory analysis. Please indicate the meaning of 10 elements, do you mean panellists?  Were the panellist trained? In case a trained panel is used the authors should include the following: training and performance evaluation of the panel. Indicate the evaluation process of the repeatability and discrimination ability of panellists.

Line 190-192. The product was evaluated after cooking at time 8d and without cooking treatment at 16 d. The results obtained for the sensory analysis of cooked and raw sausages are analysed together. In my opinion, the evaluation of the ripening time in cooked and raw samples can not be analysed together as the cooking process produce a completely different sensory profile.

Table 1. Please indicate the proportion of lean meat and fat used. Indicate the type of spices used.

Figure 1. Please correct the chart as several names are not written in English (ie. Laurel…)

This report describes a simple experimental work to check the use of KCl and commercial salt-substitute which are well known to be used as substitutes. The results showed the ability of these substances to reduce the sodium content as it is already known.  There was no elucidation of the mechanisms involved in the effect of salt substitutes in sensory and physico-chemical properties. In addition, the simple experimental work with a lot of inaccuracies (sensory methodology and experimental design not appropriate) and not very well described (long introduction and use of inappropriate references).

Author Response

Authors are very grateful to referees for their comments and suggestions that has improved the quality of the manuscript.

Referee 1

The manuscript includes 70 references. In order to see the relevance of the manuscript only those references directly related with the study should be included. In my opinion, in order to demonstrate the novelty and the results obtained, I consider that the manuscript should include a maximum number of references of 30 to 40. The authors should be aware of shorten the manuscript and only show the relevant results.

Response: the number of references were reduced according the referee indication

Lines 129-131- Please include the type of methodology used not only the reference.

Response: Done

Line 181 Sensory analysis. Please indicate the meaning of 10 elements, do you mean panellists? Were the panellist trained? In case a trained panel is used the authors should include the following: training and performance evaluation of the panel. Indicate the evaluation process of the repeatability and discrimination ability of panellists.

Response: as the reviewer well understood 10 elements mean 10 panellists’ panel. The text was rewritten so it can be more explicit. The inclusion of the training methodology does not seem to fit here, since that is not the objective of this work and because it will increase the amount of unnecessary words. However, we can inform the reviewer that the taste panel was recruited, selected and trained according to the Portuguese standards mentioned in the manuscript (mainly NP-ISO-8586-1, 2001), then they were trained and qualified for meat products. Every 12 months they do a maintenance test. In each evaluation session we don’t consider for statistical analysis the panellists that do not fulfil less than 20% of deviations (we consider a deviation if the evaluation is more than 1 point away from the median) for each attribute and use a random duplicate to check the reproducibility (if for the same sample evaluated one second time the score is similar, also equal or less than 1 point away from each other) and if more 15% deviations are found for duplicates the panellist is not considered in statistical analysis.

Line 190-192. The product was evaluated after cooking at time 8d and without cooking treatment at 16 d. The results obtained for the sensory analysis of cooked and raw sausages are analysed together. In my opinion, the evaluation of the ripening time in cooked and raw samples can not be analysed together as the cooking process produce a completely different sensory profile.

Response: 8 and 16 days ripening time sausages were not evaluated together. Their sensory evaluation was made in different sessions, statistical analysis was made separately, and results are presented in different Figures (2a and 2b).

Table 1. Please indicate the proportion of lean meat and fat used. Indicate the type of spices used.

Response: Done

Figure 1. Please correct the chart as several names are not written in English (ie. Laurel...)

Response: Done

This report describes a simple experimental work to check the use of KCl and commercial salt-substitute which are well known to be used as substitutes. The results showed the ability of these substances to reduce the sodium content as it is already known. There was no elucidation of the mechanisms involved in the effect of salt substitutes in sensory and physico-chemical properties. In addition, the simple experimental work with a lot of inaccuracies (sensory methodology and experimental design not appropriate) and not very well described (long introduction and use of inappropriate references).

Response: The ability to use the Subsalt compared to other sodium substitutes is not so known and in DOP or GPI meat sausages is a novelty. The elucidation of the mechanisms involved in the physicochemical and sensory properties were not exactly the aim of the present study. Sensory experimental methodology was conducted according to the Portuguese standards (transcribed from European standards). The effect of salt substitutes in sensory and physico-chemical properties were presented and discussed and all inaccuracies signed by the referees were corrected.

Reviewer 2 Report

Article: Effect of ripening time and NaCl replacement by other salts on the quality of Bísaro pork sausages (PGI Chouriça de Vinhais). This manuscript reports a study that evaluated the effect of NaCl partial replacement by KCl and Sub4Salt® on physicochemical properties, microbiological quality, and sensory traits of fermented sausages at different ripening times.

This study reported some interesting data, because it could improve the nutritional value of a PGI product. However, in my opinion, the paper presents some problems related to the discussion of the results. A more concise description of the experimental results and their interpretation should be drawn. The findings of physicochemical, microbiological and sensorial investigation should be discussed as similarly as possible (as number of rows). In addition, the paper needs improvements.

Specific remarks

Abstract

Line 16. Please, add partial NaCl replacement …

Line 17. Please replace “meat sausage” with “pork sausage”

Line 17. Insert a description of the salt formulations in the abstract.

Keywords. Please, delete “low-salt”. The claim “low in sodium/salt” may be used for a product containing no more than 0,12 g of sodium, or the equivalent value for salt, per 100 g.

Introduction

Line 30 and line 33. Replace reference 1 and 2 with more appropriate references.

Line 43. Please, check references. In the reference the effect of ripening time was investigated only in whole pieces.

Line 65. Please, replace “pathogenic microbial” with “pathogenic microorganism”.

Line 68. Partial replacement

Line 77. Replace “consumer’s acceptability” with “sensory traits”

Materials and methods

Line 118. Specify whether, at each replication carried out, sausages with different formulations were manufactured from the same batch of ground meat or from different batches.

Line 184. What means exterior and interior colour. Please specify.

Lines 195 and 580 and 581. Replace “tasters” with “assessors”

Line 198 ….and two ripening time

Line 196 Physicochemical and microbiological data were analysed using statistical package JMP® Pro 13.1.0………………. Statistical analysis of the sensory data……

Results and discussion

A more concise description of the experimental results and their interpretation should be drawn, especially for the paragraph 3.1. The aim of this study is to investigate the effect of ripening time and NaCl replacement by other salts on the quality of Bísaro pork sausages. To better focus the results, authors should limit, whenever it is possible, the references to studies on the partial replacement of NaCl with KCl in pork fermented sausage (i.e. ie line 43, Lines 222 - 229, line 237, lines 274 - 276, lines 354 – 359 etc ), avoiding also products from different for species (267 - 270; 273 - 274; 312 – 314 etc).

Line 212. Replace “physicochemical properties” with “physical properties”. The same in the table 2.

Line 218. Delete ‘’and the values ranged between 5.7-5.9 in days 6, 9 and 12.

Lines 240 – 243. Comparison between different meat products is not the goal of the present study. Please, delete comment and references.

Line 246. ‘’This fact is expected, because as the dry-ripening step progresses, it produces a significant and simultaneous decrease in both, moisture and aw [49]’’. The decrease is not always significant. Please, delete this sentence.

Lines 310 – 312 and lines 319 – 320. Can the authors exclude that the differences in protein and fat contents are independent from the initial batches?

Line 390. TBARs, expressed as mg MDA/kg of fat, should be compared if products differ in fat content.

Line 424. Delete ‘’all’’ before 16 days ripened sausages.

Lines 529 - 530. Please, omit this sentence or replace ‘’consumer perception’’ with ‘’consumer test’’.

Tables

Table 3. Please, complete the table with appropriate u.m.

Reference. References should be reduced

Author Response

Authors are very grateful to referees for their comments and suggestions that has improved the quality of the manuscript.

Referee 2

Line 16. Please, add partial NaCl replacement ...

Response: Done

Line 17. Please replace “meat sausage” with “pork sausage”

Response: Done

Line 17. Insert a description of the salt formulations in the abstract.

Response: Done

Keywords. Please, delete “low-salt”. The claim “low in sodium/salt” may be used for a product containing no more than 0,12 g of sodium, or the equivalent value for salt, per 100 g.

Response: Done

Line 30 and line 33. Replace reference 1 and 2 with more appropriate references.

Response: The references were eliminated. In fact this is an observation of common knowledge and does not need any reference

Line 43. Please, check references. In the reference the effect of ripening time was investigated only in whole pieces.

Response: The reference was checked and the text rewritten.

Line 65. Please, replace “pathogenic microbial” with “pathogenic microorganism”.

Response:  Done

Line 68. Partial replacement

R: Done

Line 77. Replace “consumer’s acceptability” with “sensory

traits”

Response:  Done

Line 118. Specify whether, at each replication carried out, sausages with different formulations were manufactured from the same batch of ground meat or from different batches.

Response:  Done

Line 184. What means exterior and interior colour. Please specify.

Response:  Changes were made in text

Lines 195 and 580 and 581. Replace “tasters” with “assessors”

Response:  Done

Line 198 ....and two ripening time

Response:  Done

Line 196 Physicochemical and microbiological data were analysed using statistical package JMP® Pro 13.1.0................... Statistical analysis of the sensory data......

Response:  Done

To better focus the results, authors should limit, whenever it is possible, the references to studies on the partial replacement of NaCl with KCl in pork fermented sausage (i.e. ie line 43, Lines 222 - 229, line 237, lines 274 - 276, lines 354 – 359 etc ), avoiding also products from different for species (267 - 270; 273 - 274; 312 – 314 etc).

R: The references of products from different species not pork was deleted and the text was rewritten.

Line 212. Replace “physicochemical properties” with “physical properties”. The same in the table 2.

Response:  Done

Line 218. Delete ‘’and the values ranged between 5.7-5.9 in days 6, 9 and 12.

Response:  Done

Lines 240 – 243. Comparison between different meat products is not the goal of the present study. Please, delete comment and references.

Response:  Done. The idea of comparison between different meat products was eliminated as well the comment and references.

Line 246. ‘’This fact is expected, because as the dry- ripening step progresses, it produces a significant and simultaneous decrease in both, moisture and aw [49]’’. The decrease is not always significant. Please, delete this sentence.

Response:  Done

Lines 310 – 312 and lines 319 – 320. Can the authors exclude that the differences in protein and fat contents are independent from the initial batches?

Response: Yes once all batches were produced with the same ground meat and fat as we clarified before, answering to referee comment about line 118.

Line 390. TBARs, expressed as mg MDA/kg of fat, should be compared if products differ in fat content.

Response: Comparisons were made

Line 424. Delete ‘’all’’ before 16 days ripened sausages.

Response:  Done

Lines 529 - 530. Please, omit this sentence or replace ‘’consumer perception’’ with ‘’consumer test’’.

Response:  Done

Table 3. Please, complete the table with appropriate u.m.

Response:  Done

Reference. References should be reduced

Response: The number of references was reduced

Reviewer 3 Report

This work presents information on an interesting topic: effect of NaCl replacement by others salts on the quality of dry-cure meat products. Although, several studies have been published in recent years concerning the NaCl replacement strategies, the novelty of the present work is the evaluation of NaCl replacement by other salts on physicochemical, microbiological, and sensorial quality of a traditional pork sausage (PGI Chouriça de Vinhais) of short ripening period. Although interesting, the manuscript has some issues to be solved:

Title
In my opinion, the title should be revised to better reflect the nature of the study. The main goal of the paper is to evaluate the effect of the NaCl replacement by other salts. This effect was evaluated in two different ripening time, but last subject should be not included in the title, since if this were the case, the evaluation should have been considered in more than two maturation times and the need for the evaluation of the effect of the ripening time should be better justified in the introduction. The only sentence where is referred the ripening time in the Introduction section is just the last one (“At the same time, the effect of ripening time was also assessed”).  

Introduction

Lines 39-40. Effect of reduction of NaCl content on the increase of proteolysis during ripening of dry-cured meat products and negative effect on doughy textures of these products should be indicated and some reference about this problem should be included.

Lines 73-76. In this paragraph should be justified properly how reduction/substitution of NaCl can affect the microbiological and sensorial quality of traditional ripened products with quality brands, in which NaCl is the usual ingredient, to make necessary the main goal of this paper the evaluation of effect of NaCl reduction quality of these kind of products.

Materials and methods

Lines 87- 118. The Flow chart of Chouriça de Vinhais sausage fabrication is shown in Figure 1, and the authors stated that sausages were made according to the specifications book of the PGI product. In addition, the main formulations of the four types of elaborated sausages are given in Table 1. However, more details about what spices and quantity of the spices used should be given. In addition, temperature and relative humidity specifically used should be introduced (They were the same shown in the Figure 1?). Is sugar used as ingredient in the formulation of this product? In this case should be also included.

Lines 100-103. If it is possible, more information about composition of Sub4Salt® composition should be given to the readers.  

Line 159. In microbiological analysis should justified why where tested coliforms and Escherichia coli instead a more general microbiological parameter usually determined in ripened products such as Enterobacteriaceae. In addition, why lactic acid bacteria evolution that it is the dominant microbial population in these kinds of products, was not evaluated.

Results and Discussion

Lines 305-313 and 314-325. Results of proteins and fat content should be expressed as dry matter (Table 3). Thus, discussion about increase of these parameters between days 8 and 16 stated in these paragraphs could be modified, since the reported increases could be only false effects due to the decrease of humidity content during ripening time.

As a general conclusion of the work the authors stated that the use of KCl and Sub4Salt in the reformulation of traditional Portuguese sausages showed a promising result. But, in addition, the authors have enough information to discuss what treatment NaCl + KCl,  KCl + Sub4Salt®  or NaCl + Sub4Salt could be more appropriated. This should be discussed and the end of Results and Discussion section.  

Author Response

Authors are very grateful to referees for the comments and suggestions that has improved the quality of the manuscript.

In my opinion, the title should be revised to better reflect the nature of the study. The main goal of the paper is to evaluate the effect of the NaCl replacement by other salts. This effect was evaluated in two different ripening time, but last subject should be not included in the title, since if this were the case, the evaluation should have been considered in more than two maturation times and the need for the evaluation of the effect of the ripening time should be better justified in the introduction. The only sentence where is referred the ripening time in the Introduction section is just the last one (“At the same time, the effect of ripening time was also assessed”).

Response: The title was modified according the referee suggestion.

Lines 39-40. Effect of reduction of NaCl content on the increase of proteolysis during ripening of dry-cured meat products and negative effect on doughy textures of these products should be indicated and some reference about this problem should be included.

Response: This text was modified according the other two referees.

Lines 73-76. In this paragraph should be justified properly how reduction/substitution of NaCl can affect the microbiological and sensorial quality of traditional ripened products with quality brands, in which NaCl is the usual ingredient, to make necessary the main goal of this paper the evaluation of effect of NaCl reduction quality of these kind of products.

Response: The paragraph was rewritten trying to find the referee's indications

Lines 87- 118. The Flow chart of Chouriça de Vinhais sausage fabrication is shown in Figure 1, and the authors stated that sausages were made according to the specifications book of the PGI product. In addition, the main formulations of the four types of elaborated sausages are given in Table 1. However, more details about what spices and quantity of the spices used should be given. In addition, temperature and relative humidity specifically used should be introduced (They were the same shown in the Figure 1?). Is sugar used as ingredient in the formulation of this product? In this case should be also included.

Response: The Flow chart in figure was referenced to PGI specifications (Commission Regulation (EC) No 1265/98 of 18 June 1998). Spices on Table 1 were detailed. Sugar is not an ingredient used. All fabrication process of sausages followed the temperature and moisture and all procedures indicated in Figure 1. The only ingredient that varies was the quantities and type of salt. Changes were made in Figure and Table 1.

Lines 100-103. If it is possible, more information about composition of Sub4Salt® composition should be given to the readers.

Response: As a product available on the market, we believe we have provided the necessary information

Line 159. In microbiological analysis should justified why where tested coliforms and Escherichia coli instead a more general microbiological parameter usually determined in ripened products such as Enterobacteriaceae. In addition, why lactic acid bacteria evolution that it is the dominant microbial population in these kinds of products, was not evaluated.

Response: In our study we determined the presence of Salmonella sp. and the counts of moulds and yeasts, aerobic mesophilic bacteria, total coliforms, Staphylococcus aureus, Escherichia coli and sulphite-reducing clostridia spores. These parameters were chosen for two reasons: the first, is because these are the most relevant when the concern is the microbiological quality of the product, the second is because our laboratory is certified to develop those assays by the Portuguese Regulatory Entity. The authors decided not to assess the quantity of lactic acid bacteria because these are not microbial hazards on cured meat but are, instead, protective agents that produce compounds with antimicrobial activity, contributing to the exclusion of meat pathogens. The determination of lactic acid bacteria was out of the scope of our objective of “pathogens control”; however, we will undoubtedly take reviewer’s comment into account in our future works and include that determination (after proper certification).

Lines 305-313 and 314-325. Results of proteins and fat content should be expressed as dry matter (Table 3). Thus, discussion about increase of these parameters between days 8 and 16 stated in these paragraphs could be modified, since the reported increases could be only false effects due to the decrease of humidity content during ripening time.

Response: The results were expressed in the form of providing the necessary comparisons with other studies. Naturally, we believe that the reduction of humidity increases the levels of protein and fat, which are expressed in% in the specifications of PGI products (Commission Regulation (EC) No 1265/98 of 18 June 1998).

As a general conclusion of the work the authors stated that the use of KCl and Sub4Salt in the reformulation of traditional Portuguese sausages showed a promising result. But, in addition, the authors have enough information to discuss what treatment NaCl + KCl, KCl + Sub4Salt® or NaCl + Sub4Salt could be more appropriated. This should be discussed and the end of Results and Discussion section.

Response: According to the available results, we believe to have provided adequate information for the reduction or even replacement of Na in meat sausages. Anyway, a global analysis at the end of results and discussion section was included according the referee’s suggestion.

Round 2

Reviewer 1 Report

I indicated my opinion about the manuscript in the first version. It is merely a technological study and not a scientific study. So I am not changing my first opinion as any improvement in the scientific quality has been done.

Author Response

I apologize but after 37 years working in animal science it is hard for me to understand and identify the frontier between a technological and a scientific study to classify the present manuscript as “merely a technological and not scientific study”. Anyway, having accepted all suggestions and responded to all comments made by the referee in review 1 we are surprised by the referee’s opinion that the work does not deserve to be published because we did not introduce any scientific improvement when all the comments made by him were answered and the suggestions adopted. However, I am grateful for your important comments in the first review that   have improved the quality of the manuscript.

Reviewer 2 Report

Suggestions have been accepted by authors and manuscript has been improved

Author Response

Referee 2:

Suggestions have been accepted by authors and manuscript has been improved.

Response: As no other specific comment or suggestions has been made by the referee, we believe the manuscript is in conditions to be accept.

Reviewer 3 Report

Authors have followed most of the recommendation and suggestion of the Reviewers. Title, discussion, and general conclusion have been modified following recommendation of the Reviewers.

Author Response

Referee 3:

Authors have followed most of the recommendation and suggestion of the Reviewers. Title, discussion, and general conclusion have been modified following recommendation of the Reviewers.

Response: As no other specific comment or suggestions has been made by the referee, we believe the manuscript is in conditions to be accept.